# The Combination of Gold and Silver Food Nanoparticles with Gluten Peptides Alters the Autophagic Pathway in Intestinal Crypt-like Cells

**DOI:** 10.3390/ijms241713040

**Published:** 2023-08-22

**Authors:** Clara Mancuso, Eric Tremblay, Elisa Gnodi, Steve Jean, Jean-François Beaulieu, Donatella Barisani

**Affiliations:** 1School of Medicine and Surgery, University of Milano-Bicocca, 20900 Monza, Italy; clara.mancuso@unimib.it (C.M.); elisa.gnodi@unimib.it (E.G.); 2Laboratory of Intestinal Physiopathology, Faculty of Medicine and Health Sciences, Université de Sherbrooke, Sherbrooke, QC J1H 5H4, Canada; eric.tremblay@usherbrooke.ca (E.T.); jean-francois.beaulieu@usherbrooke.ca (J.-F.B.); 3Department of Immunology and Cell Biology, Faculty of Medicine and Health Sciences, Université de Sherbrooke, Sherbrooke, QC J1H 5H4, Canada; steve.jean@usherbrooke.ca

**Keywords:** food additives, celiac disease, autophagy, gluten peptides, dietary metallic nanoparticles, lysosome dysfunction

## Abstract

Metallic nanoparticles (mNPs) are widely used as food additives and can interact with gliadin triggering an immune response, but evaluation of the effects on crypts, hypertrophic in celiac subjects, is still lacking. This study evaluated the effects of gold and silver mNPs in combination with gliadin on crypt-like cells (HIEC-6). Transmission electron microscopy (TEM) was used to evaluate gliadin-mNP aggregates in cells. Western blot and immunofluorescence analysis assessed autophagy-related molecule levels (p62, LC3, beclin-1, EGFR). Lysosome functionality was tested with acridine orange (AO) and Magic Red assays. TEM identified an increase in autophagic vacuoles after exposure to gliadin + mNPs, as also detected by significant increments in LC3-II and p62 expression. Immunofluorescence confirmed the presence of mature autophagosomes, showing LC3 and p62 colocalization, indicating an altered autophagic flux, further assessed with EGFR degradation, AO and Magic Red assays. The results showed a significant reduction in lysosomal enzyme activity and a modest reduction in acidity. Thus, gliadin + mNPs can block the autophagic flux inducing a lysosomal defect. The alteration of this pathway, essential for cell function, can lead to cell damage and death. The potential effects of this copresence in food should be further characterized to avoid a negative impact on celiac disease subjects.

## 1. Introduction

Celiac disease (CeD) is a multifactorial autoimmune disorder that develops in subjects carrying the specific genetic HLA-DQ2 and/or -DQ8 background, and it is triggered by the ingestion of gluten. Gluten is composed of a heterogeneous complex of proteins rich in glutamine and proline and is contained in several types of cereals such as wheat, rye and barley [1]. Even though gluten is a common component of the Western diet, and the HLA-DQ2 polymorphism is present in about 30% of the general population [2], CeD affects only 2–5% of these subjects, suggesting the presence of other environmental factors contributing to its etiology. In this sense, several efforts have recently been made to identify possible new ones, evaluating viral infections [3], increased intestinal permeability [4], variations in duodenal microbiota [5] and the growing use of food additives in the industry [6,7].

In this context, the appearance of metallic nanoparticles (mNPs) in the food chain seems particularly alarming, since their ingestion has been linked to intestinal permeability alterations, cytotoxic effects and immune system impairment [8], as well as to inflammatory bowel disease and colon neoplastic lesions [9,10]. However, the potential involvement of dietary mNPs in the increasing incidence of CeD is still poorly documented. mNPs present a typical dimension of 1–100 nm or higher, and their use in food is regulated by the European Food Safety Authority (EFSA) in Europe and the Food and Drug Administration in the US [11]. Their applications are numerous, including the agro/feed/food chain [12,13,14,15], in which they are used as dyes or additives to improve organoleptic properties, as antimicrobial agents to preserve foods or improve crop growth and to increase the bioavailability of nutritional supplements. Silver (AgNPs) and gold (AuNPs) metal nanoparticles are among the most commonly used mNPs in food, and their food additive codes in Europe are E174 and E175, respectively. If the use of AgNPs in food packaging for their antimicrobial properties does not raise many concerns, since it was demonstrated that the release of Ag ions is limited, the use of E174 and E175 as food colorants still needs to be fully characterized [16,17,18]. The main concerns about their presence in food include potential cytotoxicity, the induction of oxidative stress and dysbiosis for E174 and, mainly, the potential accumulation in tissues for E175, but more data are needed. Moreover, the EFSA has estimated that their daily intake is higher in children than in adults since they are mainly found in sweets and ultraprocessed food (estimated intake in toddlers: E174, 0.003–0.08 mg/kg/day; E175, 0.01–0.26 μg/kg/day), increasing the concerns about their potential effects on a growing intestine [17,18]. Although data about their toxicity on differentiated intestinal cells are increasing [7,19,20,21,22], studies regarding their potential effects on intestinal crypt-like cells have not been performed yet. In addition, mNPs can interact with food components forming agglomerates and aggregates, and many studies use different mNP concentrations, making it difficult to understand their main effects [23].

With these premises, the possible combined effects of digested gliadin (the main immunogenic gluten component) with Ag- or Au-NPs were analyzed on the normal human intestinal epithelial crypt-like (HIEC-6) cell line [24], a cell model that has been used for the evaluation of the mechanisms underlying intestinal stemness [25], differentiation [26] and cytotoxicity [27]. In particular, we focused on the potential effects of mNP-gliadin complexes on the macroautophagic process (hereafter referred to as autophagy), essential for cellular well-being. Briefly, various membrane reservoirs and factors are involved in the nucleation and expansion of the phagophore membrane, among which is Beclin-1. The small ubiquitin-like Atg8/LC3 protein family is implicated in the phagophore closure and in its maturation into the autophagosome; this process is characterized by the phosphatidylethanolamine (PE) conjugation of the microtubule-associated proteins 1A/1B light chain 3B (LC3-I) to form LC3-II protein. Finally, mature autophagosomes fuse with the lysosomes, where the sequestered materials are degraded, and most of their components are recycled [28]. Thus, autophagy is an important intracellular pathway involving the sequestration of different cargos, such as damaged organelles, aggregated proteins and pathogens, into double-membrane autophagic vacuoles (AVs) and their consequent lysosomal-mediated degradation [29]. 

It has been recently reported that the autophagic pathway may have a role in mNP toxicity and, even more interesting, in CeD pathogenesis, since the accumulation of mNPs or gluten peptides in autophagic vesicles seems to alter their proper functionality [8,30,31,32]. These impaired processes could be important not only in fully differentiated cells but also in all the different types of enterocytes, since Groulx et al. [33] showed that autophagy is active in intestinal proliferative/undifferentiated cell populations (e.g., the HIEC-6 cell line). 

The data herein presented demonstrate the presence of autophagic alterations in response to mNP and gliadin exposure, which also involves the lysosomal function, as assessed by transmission electronic microscopy (TEM), the evaluation of the proteins involved in autophagosome formation and lysosomal assays.

## 2. Results

### 2.1. TEM Visualization of mNP-PT Complexes Taken up by HIEC-6

For the experiments, we selected HIEC-6 cells, which were originally isolated from the intestine of a human fetus. As previously reported, these cell express specific intestinal keratins 8, 18 and 21, as well as aminopeptidase N and dipeptidylpeptidase IV [24], but lack Cdx2 and HNF-1a like undifferentiated crypt cells [26]. Similarly to what we previously reported in CaCo2 cells [34], we selected a concentration of 25 μg/mL for AuNPs and 5 μg/mL for AgNPs since they were not inducing morphological changes in HIEC-6 cells (Appendix A). Previous TEM and spectrophotometric analysis performed in our laboratories demonstrated an increased aggregation of mNPs in the presence of digested gliadin (PT, 1 mg/mL) [34], a fact that could be important in determining their intracellular fate. Thus, HIEC-6 cells were treated for 24 h with mNPs and/or digested gliadin and then processed for TEM visualization. Controls showed a typical appearance of undifferentiated cells with a poorly organized cytoplasm and only a few intracellular vacuoles (Figure 1A,B). Cells treated with either PT (Figure 1C,D) or mNPs (Appendix A) alone displayed a similar appearance as the controls. However, an important increase in the number of AVs, including both immature and mature autophagosomes, was observed in cells treated with the combination of mNPs and PT (Figure 1E–H). These results were similar to those obtained treating the cells with 50 nM bafilomycin (BafA), a compound able to neutralize the lysosomal pH through the inactivation of the proton pump V-ATPases but also able to block the autophagy flux via the inhibition of the autophagosome–lysosome fusion, thus inducing AV accumulation into the cells (Appendix A) [35]. Moreover, a large amount of aggregated NPs were observed inside the cells after stimulations with the combination of PT and mNPs, particularly AuNPs (Figure 1E,F), and were strictly associated with the intracellular vacuoles. On the contrary, in cells stimulated with mNPs alone, a low percentage of vacuoles was seen, as well as a reduced number of aggregated or isolated mNPs in the endosomal vesicles (Appendix A). Although we did not perform a physical analysis of the material present in the vesicles, the differences observed comparing the treatment with NPs alone or associated with gliadin suggest that there is an interaction between these two components that increases their ability to affect autophagy.

### 2.2. Molecular Markers Indicate Autophagy Alterations

#### 2.2.1. Western Blot

LC3 is the most used molecular marker for the evaluation of autophagy [36]. It exists in two separated forms, LC3-I (16–18 kDa) and its PE-conjugated form LC3-II, which although larger in mass, shows faster electrophoretic mobility in SDS-PAGE gels. Particularly, LC3-II is a protein marker associated with complete autophagosomes. Therefore, measuring the transformation of LC3-I to LC3-II can give information about autophagy alterations [37]. Although no alterations in LC3-I were observed under the tested conditions (Figure 2A), a significant increase in LC3-II protein levels was seen after stimulation with both Au- and Ag-NPs when combined with PT versus both the untreated cells (*p* < 0.01 and *p* < 0.05, respectively) and the PT treatment alone (*p* < 0.01 and *p* < 0.05, respectively) (Figure 2B). 

In addition, a slight increment in the LC3II/I ratio was observed after stimulations with the combination of AgNPs and PT (Figure 2C). To further characterize a possible alteration in the autophagic flux, we also evaluated the p62 protein level. p62 is a multiadaptor protein interacting with both polyubiquitinated proteins and LC3-II on autophagosomes for engulfment. At the same time, it remains incorporated into the autophagosome and is degraded only after fusion with the lysosome, thus serving as an index of autophagic degradation. A significant increase in its protein level was observed after stimulation with both the Au- and Ag-NPs complexes as compared to controls (*p* < 0.001) (Figure 2D). Since other signals may induce an increase in p62 through an induction in transcription, we also evaluated its mRNA level through qPCR (SQSTM1 gene), but only a slight increase in its transcript level was seen with the same stimulations (Appendix A), thus confirming that the higher p62 protein level was related to an altered autophagy flux. 

For further confirmation, we then measured levels of ATG6/Beclin-1, a component involved in the initial formation of AVs, and no increase was observed at both the mRNA (Appendix A) and protein level (Appendix A), indicating that mNP + PT stimulations did not induce an increase in AV formations. Considering the previously obtained results on the p62 protein, we thus hypothesized that mNP plus PT treatments may increase AVs by blocking their degradation. 

#### 2.2.2. Immunofluorescence

Further analyses were carried out performing indirect immunofluorescence of autophagy markers, staining cells with both anti-p62 and anti-LC3 antibodies (Figure 3). 

Initially, for both the proteins evaluated, a diffuse pattern was observed in the negative control (untreated cells; Figure 3A) and under stimulation with PT (Figure 3B) or with mNPs alone (Appendix A). However, a more punctate pattern was seen in cells stimulated with mNPs and PT (Figure 3C,D), even if the average number of puncta per cell was less than in the positive 50 nM BafA control (Figure 3E), as expected from the TEM observations. However, the anti-LC3 antibody identifies both the LC3-I and LC3-II forms, but LC3 and p62 colocalize only in mature autophagosomes; therefore, overlaying the signal with p62 staining would more precisely indicate mature autophagosomes. Thus, the fraction of the LC3 signal overlapping the p62 one was quantified, and a higher colocalization of p62 and LC3 signals was observed following stimulation with the combination of mNPs and PT compared both to the control and to PT alone (Figure 3). In particular, the colocalization was significantly higher after AgNP + PT and BafA treatments (*p* < 0.05 vs. ctr) (Figure 4), confirming the accumulation of AVs in the cells.

### 2.3. The Accumulation of AVs Is Not Due to Autophagy Induction

Normally, autophagosome accumulation may be linked to an increased autophagy induction and/or to a block in the autophagy flux. This last mechanism can be induced by a block in the trafficking of AVs to lysosomes and/or by lysosomal dysfunctions, leading to a reduction in AV degradation. Therefore, to assess whether the observed alterations in the autophagy pathway involve a block in the flux or an increase in AV biogenesis, cells were incubated with mNPs + PT, with and without BafA. 

In fact, the difference in the amount of LC3-II and p62 proteins, in the presence and absence of lysosomal inhibitors, can be used to examine autophagy flux. Higher protein levels with a specific treatment added with BafA, compared to the inhibitor alone, may indicate that the treatment increases AV biogenesis; on the contrary, if the specific treatment combined with BafA increases the protein level comparably to BafA alone, this may indicate a partial block in the autophagy flux [32]. As expected, stimulations with the positive control BafA induced an increase in both LC3-II (Figure 5C) and p62 proteins (*p* = 0.06) (Figure 5D), as well as a significant increase in the LC3-II/I ratio (*p* < 0.5) versus controls (Figure 5B).

Although a significantly higher level of these proteins (LC3-II; ratio LC3-II/I; p62) was seen after the combination of both AuNPs + PT and AgNPs + PT with BafA (*p* < 0.01 and *p* < 0.05; *p* < 0.05 and *p* < 0.01; *p* < 0.01 and *p* < 0.001; respectively) (Figure 5B–D, respectively), this increment was not significant compared to the BafA treatment alone, suggesting that mNP-PT complexes alter autophagy by blocking its flux. 

### 2.4. EGFR Degradation Is Delayed by mNP + PT Complexes

Since EGFR is degraded through AVs in the presence of its ligand, EGF, a delay in its degradation upon EGF exposure could indicate defects in the AV flux. Therefore, to further confirm a block in the autophagy flux, we evaluated EGFR degradation after stimulations with either BafA or the combinations of PT plus Au- or Ag-NPs. As expected, we observed a significant reduction in the EGFR levels over time in the untreated cells exposed to EGF (Figure 6). 

A delay in EGFR degradation was seen after stimulation with either PT or the mNPs alone, whereas an impairment in EGFR degradation similar to the one induced by BafA was caused by the combination of AuNPs + PT and AgNPs + PT, particularly after 60 min from the EGF pulsing (Figure 6). This confirms that mNPs + PT can alter the autophagy flux, either impairing vesicle trafficking or altering the lysosomal function. Thus, we decided to evaluate lysosome acidity and functionality.

### 2.5. Lysosomes Are Involved in the Impairment of the AV Flux

#### 2.5.1. Acridine Orange Staining

To evaluate if defects in lysosomes are responsible for the observed AV alterations, both their acidity and the enzymatic functions were analyzed. Initially, lysosomal acidity was assessed using acridine orange (AO) staining. This is a fluorophore that dimerizes when it accumulates in an acidic environment, which causes a shift from green to red in its fluorescent emission. Consequently, the red emission is representative of the intracellular acidic lysosomal and autophagic vesicles, whereas the green signal stains the diffuse non-autophagic compartment. Since the signal may be affected by cell density and size, a red/green ratio was used to normalize the acidic AO signal (as described by SenthilKumar et al. and Thomé et al.) [38,39]. Therefore, alterations in lysosome acidity, as induced in our experiments by the positive control BafA 50 nM (*p* < 0.01 vs. ctr) (Figure 7A), were detected by a reduced AO red/green fluorescence. 

Interestingly, a 30% reduction in the signal was observed after stimulations with mNPs plus PT (AuNPs + PT 0.98 ± 0.05; AgNPs + PT 0.97 ± 0.14; medium 1.47 ± 0.04; means ± SD), although it did not reach significance (Figure 7A), indicating a mild affection of lysosome pH. 

#### 2.5.2. Magic Red Assay

To fully evaluate lysosome functionality, it is also important to consider their enzymatic activity. To do so, we employed a Magic Red (MR) assay, which measures the enzymatic activity of cathepsin B, one of the major cysteine lysosomal proteases. In this case, a reduction in the MR fluorescent signal means a decrease in the lysosomal functionality. For this assay, we used leupeptin as a positive control, a potent inhibitor of cathepsin B in vitro. As expected, a significant reduction in the MR signal was observed after stimulation with leupeptin at a concentration of 20 uM (*p* < 0.0001) (Figure 7B). Similarly, the MR signal was significantly reduced by treatments with the combination of both Au- and Ag-NPs with PT (*p* < 0.001 and *p* < 0.0001, respectively) (Figure 7B). This confirmed an impairment in lysosome functionality caused by cell exposure to mNP + PT complexes, which could be the mechanism responsible for the observed defects in the AV flux. 

## 3. Discussion

It is quite clear nowadays that the crosstalk between autophagy and immune-related signaling is one of the main defense mechanisms of cells, a fact which has led to the rising interest in these pathways when studying chronic diseases with an inflammatory and autoimmune background [40]. The uptake of external threats through the autophagic route is indeed a double-edged sword, since it is one of the most effective ways through which cells can dispose of them, but it can itself become problematic when it loses its efficiency. In fact, the engulfment of the autophagic pathway can lead to the accumulation of autophagolysosomes, triggering a pathway potentially even more toxic for the cell [8]. It has already been demonstrated that mNPs can be taken up through this mechanism, interfering with it and leading to their accumulation inside the cells. In turn, this could lead to the formation of ROS and cause a consequent cell toxicity [8]. 

For this study, we employed HIEC-6 cells, a nontumoral cell line derived from fetal intestine which maintains the characteristics of intestinal crypt cells, thus recapitulating the features of the cells that, in celiac disease, present an increased proliferation, due to the destruction of the villi and the deepening of the crypts [24,26]. The similarity with the crypt cells also regards authophagy, which is more abundant in undifferentiated HIEC-6 cells as well as in the cells present in the deeper part of the crypts [33]. Given the importance of these processes, we wanted to assess whether mNPs, alone or in combination with gliadin, were entering the cells and inducing an alteration of the cellular pathways. Indeed, we demonstrated that HIEC-6 intestinal cells can take up mNPs through endocytic vesicles, as clearly visible in the cell cytoplasm. This accumulation was even more evident when mNPs were combined with PT gliadin, the part of gluten responsible for the immunogenic reaction in celiac subjects, suggesting that the combination of mNPs + PT can affect more profoundly the autophagic route. The presence of a stronger effect with combination treatments had already been reported in our previous work on CaCo2 cells and biopsies from CeD subjects on a gluten-free diet, compromising the barrier integrity in the former and inducing inflammatory cytokines in the latter. In the same study, we also observed that PT could interact with the mNPs, as assessed by a change in their UV-vis spectra and TEM analysis [34]. This aspect is quite important, since mNPs have potential interactions with the food matrix and digestion fluids, which could induce mNP aggregation and the formation of a protein corona, thus altering mNP properties and their possible effects on the intestine, including the way they are handled in the phagocytosis process. Different works have tried to mimic a proper digestion system, analyzing how these processes changed the mNP toxicity and reactivity, but results were sometimes discordant, mainly because of the different settings and nonstandardized systems in the different studies [21,23,41]. 

We observed that mNPs + PT can cause the engulfment of the autophagic pathway, with the accumulation of mature autophagosomes in HIEC-6 cells. A crucial point in understanding the mechanisms underlying this accumulation is determining whether it is caused by an induction of the autophagic pathway itself or rather by its blockade. The potential induction of autophagy by mNPs has been investigated in the treatment of cancer, since it seems that silver nanostructures can induce a protective autophagy in HepG2 cells, but AgNPs could also cause lysosomal defects in PC-3 prostate cancer cells, suggesting that mNPs could impact on autophagy in different ways [42,43]. In the present work, our results suggest that the accumulation of AVs is mainly due to a blockade in the autophagic route rather than its induction, since the increase in the expression of LC3-II and p62 was only observed at the protein and not the mRNA level. Even if p62 is used as an index of autophagic degradation, different mechanisms could result in its increase [44], including the accumulation of misfolded proteins, a process that could be augmented by mNPs [44]. However, the eventuality of an induction in the autophagic processes was ruled out, since the contextual stimulation with BafA + the different treatments did not cause a significant increase in the protein levels of p62 and LC3 compared to BafA alone. On the contrary, these results indicate that the observed phenomena could be mainly due to a partial block in the autophagic flux [37], and the observed impairment in EGFR degradation was a further confirmation. Barone et al. previously demonstrated that the gliadin-derived peptides (in particular, peptide 31–43) can be taken up by endocytic vesicles and delay vesicle trafficking in intestinal cells, especially interfering with the degradation of EGFR [45,46]. This could, in turn, prolong EGFR activation and increase cell proliferation in crypt enterocytes, which is one of the features of the mucosal lesion typical of celiac disease. In HIEC-6 cells, we observed not only a delay in the degradation of EGFR caused by PT, but its combination with mNPs could totally impair its degradation, results similar to those observed after treatment with the positive control, BafA. Considering that HIEC-6 cells are, by definition, epithelial crypt-like cells, these effects should not be underestimated, and the downstream impact of the persistence of EGFR in the cell in the presence of mNPs + PT could be an interesting point to investigate in the near future. However, the lysosomal dysfunction observed in our experiments explained the block in autophagy. In fact, even if the lysosomal acidity was only partially disrupted, the enzymatic activity of cathepsin B was impaired by mNPs + PT. 

Interestingly, the alteration of lysosomal function can also induce defects in the intracellular trafficking, as has been described for sphyngolipids [47]. Thus, it is possible that other additional mechanisms may concur to the autophagy flux blockage, such as those altering the cytoskeleton, which has an essential role in this pathway, being involved in the delivery of both autophagosomes and endosomes to the lysosomes for their successive fusion [48,49]. Currently, we cannot exclude a partial effect of either gliadin or mNPs on this component. Different works have demonstrated that gliadin peptides can alter the actin cytoskeleton, as well as delay the endocytic trafficking [45,50], but these data were obtained employing CaCo2 cells, which present different characteristics compared to HIEC-6. It has also been shown that AgNPs at low concentrations might interfere with cytoskeleton dynamics in both adult neuronal progenitor cells and in colon cancer cells [51,52], but even in this case, data were obtained in cells either belonging to a different tissue or derived from a cancerous one. 

It is interesting to note that in a recent paper by Xu et al., the effects of polistirene NPs (PS-NPs) on both cancerous and HIEC-6 cell lines were evaluated [53], obtaining results similar to those described here. In fact, it was observed that these kinds of NPs could be taken up both by cancerous and HIEC-6 cells through the autophagic pathway, accumulating in the autophagosomes. Additionally, the authors detected an impairment in the autophagic degradation due to lysosome dysfunction in cells but also an increase in the expression of LC3 and p62 protein in vivo, in mouse colon, after the oral administration of PS-NPs. These data suggest that even if the core material is different, the autophagic mechanisms impaired are similar and need to be further dissected. 

The evaluation of the impairment of this pathway could also be very important considering the large use of dietary mNPs in prepared food, which is becoming increasingly consumed, especially in Western countries. Moreover, it must be remembered that the food color E171 has recently been advised as “not safe” by the European Food Safety Authority as a food additive for its content of TiO_2_ mNPs. For this reason, these initial results on the impact of gold and silver mNPs in combination with PT on HIEC-6 cells can be a starting point for the further evaluation of the risks of mNP use in food additives, especially in the population of subjects predisposed to develop CeD or other intestinal diseases with an inflammatory condition.

## 4. Materials and Methods

### 4.1. Enzymatic Digested Gliadin

Commercial wheat gliadin (Sigma-Aldrich, St. Louis, MO, USA) was subjected to pepsin (0.1 M HCl, pH 1.8) (Sigma-Aldrich, St. Louis, MO, USA) and then trypsin (pH 7.8) (Sigma-Aldrich, St. Louis, MO, USA) digestion at 37 °C for 4 h with vigorous agitation. The pH was adjusted to 4.5 resulting in a precipitate, which was removed by centrifugation. N-tosyl-l-phenylalanine chlorome-thyl ketone and N- α-tosyl-l-lysine chloromethyl ketone hydrochloride (Sigma-Aldrich, St. Louis, MO, USA) were used to inhibit the residual enzymatic activity. The solution was dialyzed, filtered and lyophilized. The powder was dissolved in sterile water at 50 mg/mL, aliquoted and stored at −80 °C.

### 4.2. Nanoparticles

The commercial AgNPs (40 nm diameter) were purchased from Sigma-Aldrich (St. Louis, MO, USA; Cat #730807, 0.02 mg/mL in aqueous buffer, with sodium citrate as stabilizer); AuNPs (15 nm diameter) were obtained from Cytodiagnostics (Burlington, ON, Canada; Cat #CG-15-XX, supplied in 0.1 mg/mL citrate buffer supplemented with a proprietary stabilizing solution). The purchased nanoparticles were stored at 4 °C (according to the manufacturer’s instructions) until use. Nanoparticles, alone or in combination with gliadin, were analyzed by TEM and UV-Vis spectra as previously described [34]. Briefly, TEM confirmed the average size of the particles, i.e., 40 nm for silver and 15 nm for gold NPs, as well as their interaction with gliadin which caused a shift in the UV-vis spectra [34]. 

### 4.3. HIEC-6 Cell Culture

The crypt-like human intestinal epithelial cells (HIEC-6), as previously described [24], were grown in 75T culture flasks (with filter cap) with OptiMEM (GibcoTM—Thermo Fisher Scientific, Waltham, MA, USA) supplemented with 4% FBS, 1% Hepes, 1% glutamine (all from EuroClone^®^, Pero, Italy) and 5 ng/mL hrEGF (GibcoTM—Thermo Fisher Scientific, Waltham, MA, USA). To maintain the cell’s characteristics, all the experiments were performed with cells before the 18th passage.

For the described experiments, cells were seeded with a density of 150 × 10^3^/cm^2^ in 6 well plates. After reaching confluence, they were stimulated with peptic tryptic-digested gliadin (PT, 1 mg/mL) combined or not with AgNPs (5 μg/mL) and AuNPs (25 μg/mL), BafA (Sigma-Aldrich, St. Louis, MO, USA) 50 nM, leupeptin (Sigma-Aldrich, St. Louis, MO, USA) 20µM or left with medium alone (controls). Cells were incubated for 24 h with the different treatments, except for the stimulations with BafA or leupeptin which were added 2 h before the expiration of the other treatments. In order to maintain the necessary elements of the culture medium and avoid to concentrate the mNPs, procedure which carried the risk of aggregation, complete medium was concentrated 4× with SpeedVAC (Thermo Fisher Scientific, Waltham, MA, USA) at 30 °C for 1 h. The different treatments (PT, AuNPs, AgNPs) were then added to the 4× medium, followed by PBS 1× (EuroClone^®^, Pero MI, Italy), to reach a final volume of 750 uL/well and then were thoroughly mixed, just before dispensing them dropwise onto the confluent cells.

### 4.4. Transmission Electron Microscope (TEM)

After stimulations, cells were rinsed with phosphate-buffered saline (PBS) (EuroClone^®^, Pero, Italy), fixed for 30 min with freshly prepared 1.5% glutaraldehyde sodium cacodylate (0.1 M, pH 7.4) at room temperature and overnight with 2.5% glutaraldehyde sodium cacodylate (0.1 M, pH 7.4) at 4 °C. Subsequently, cells were washed twice with sodium cacodylate (0.1 M, pH 7.4) and postfixed in uranyl acetate 1% overnight at 4 °C (in darkness). Then, they were rinsed twice with distilled water and dehydrated with an increasing percentage of EtOH solutions (from 40% to 100%). Cells were then Epon-embedded and positioned on carbon-coated copper grids. Staining with uranyl acetate 2% was performed, and pictures were taken using a HITACHI H-7500 transmission electron microscope (Hitachi Ltd., Tokyo, Japan).

### 4.5. Protein Extraction and Western Blot (WB)

Total proteins were extracted in Ripa buffer, and 30 ug of proteins was loaded for Western blot analyses, performed on SDS-PAGE gels under denaturing conditions. For the analysis of Beclin1, LC3 and p62 proteins, 12% bis-acrylamide gels were employed, whereas EGFR proteins were separated on Nu-PAGE 4–12% Bis-Tris Gel (Invitrogen™, Thermo Fisher Scientific, Waltham, MA, USA). Proteins were then electrotransferred onto a nitrocellulose membrane, and nonspecific protein binding was blocked using 10% Blotto-0.1% Tween PBS or 5% BSA-0.1% Tween PBS, respectively. Membranes were incubated with primary antibodies (anti-LC3, L8918, Sigma Aldrich; anti-p62, 610832, BD Biosciences; anti-Beclin1, ab207612, Abcam; anti-EGFR, #2232, Cell Signaling; anti-β-actin A2066 Sigma-Aldrich) diluted 1:1000 in the blocking solution, overnight at 4 °C. Horseradish peroxidase-conjugated secondary antibodies (antimouse LNA931V, GE Healthcare, antirabbit LNA934V, GE Healthcare) were used to detect the primary antibodies, and blot density was quantified with ImageJ™ software (NIH, Bethesda, MD, USA). Protein band quantifications were normalized on the total protein band at 48 KDa (Ponceau Red band) or on β-actin, according to the most appropriate technique for the experiment.

### 4.6. Indirect Immunofluorescence

For immunofluorescence, HIEC-6 cells were fixed in MeOH (Sigma-Aldrich, St. Louis, MO, USA) for 10 min at −20 °C, and nonspecific sites were blocked for 1 h at room temperature with 10% Blotto-PBS (pH 7.4). Primary antibodies were diluted 1:1000 in a blocking solution containing 0.05% azide and incubated overnight at 4 °C. The secondary antibodies were used at 1:400 dilution in the blocking solution and incubated at room temperature for 1 h. Nuclei were stained with DAPI 1:3000 (Sigma-Aldrich, St. Louis, MO, USA) in PBS for 3 min at room temperature. The slides were visualized with a DMRXA microscope (Leica, Nussloch, Germany) equipped for epifluorescence and digital imaging (RTE/CCD Y/Hz-I300 cooled camera). Images were acquired using MetaMorph software (Universal Imaging Corporation, New York, NY, USA) with 20× and 40× objectives. Fluorescence was quantified with ImageJ™ software, JACoP plugin (NIH, Bethesda, MD, USA).

### 4.7. RNA Extraction, RT and Quantitative PCR

HIEC-6 cells were lysed with TRIzol (Invitrogen, Burlington, ON, Canada), RNA extracted according to the manufacturer’s protocol and treated with DNAse (Invitrogen™, Thermo Fisher Scientific™, Waltham, MA, USA). Retrotranscription was performed using SuperScriptII (Invitrogen™, Thermo Fisher Scientific™, Waltham, MA, USA) according to the manufacturer’s instructions. qPCR was performed using an Mx3000P system (Stratagene, Cedar Creek, TX, USA) and Brilliant II SYBR Green QPCR Master Mix (Stratagene). Differences in gene expression were evaluated by comparing untreated (medium) cells vs. stimulated ones. All data were normalized using HPRT1 and PPIA genes, and relative expression was assessed by the 2^−ΔΔCt^ method using an external control. All samples were run in duplicate, and the nontemplate control did not show an amplification product.

### 4.8. Degradation of the Epidermal Growth Factor Receptor (EGFR)

HIEC-6 cells were exposed to 20 ng/mL of hrEGF (Gibco™, Thermo Fisher Scientific, Whaltham, MA, USA) for 0 to 60 min, after being pretreated with Ag- and Au-NPs ± PT for 24 h, BafA 50 nM for 2 h or left with medium alone. Cells were then lysed, protein extracted and resolved by WB analyses as previously described.

### 4.9. Acridine Orange (AO) and Magic Red Assay (MR)

Cells were seeded with a density of 62 × 10^3^ cm^2^, and after reaching confluence, they were stimulated with Ag- and Au-NPs ± PT for 24 h, BafA 50 nM for 2 h or left with medium alone. Acridine orange (Bio-Rad, Hercules, CA, USA) at the concentration of 1 µg/mL was then added to the cells and incubated for 30 min. Cells were washed with PBS, and fluorescence signals (both red and green) were detected with a plate reader (excitation 480 nm; emission 620 nm for red, 550 nm for green). Data are presented as red/green signals ratio as previously described [38,39]. For the Magic Red assay, HIEC-6 cells were exposed to treatments for 24 h, leupeptin 20 nM for 2 h or left with medium alone. Magic Red (Bio-Rad, Hercules, CA, USA) was then added to the cells and incubated for 30 min. Cells were washed with PBS, and fluorescence signals were detected with a plate reader (excitation 595 nm; emission 635). The unlabeled cells and the noncellular test, run as negative controls, showed null or negligible signals.

### 4.10. Statistical Analyses

Independent experiments were repeated at least 3 times in duplicate. Due to the high number of tested conditions, an ANOVA multiple comparison test (Kruskal–Wallis statistical test) was performed. The evaluation of outliers was performed using the Grubbs’ and ROUT tests. Statistical analyses were performed with SYSTAT software (SPSS, Chicago, IL, USA).

## Figures and Tables

**Figure 1 ijms-24-13040-f001:**
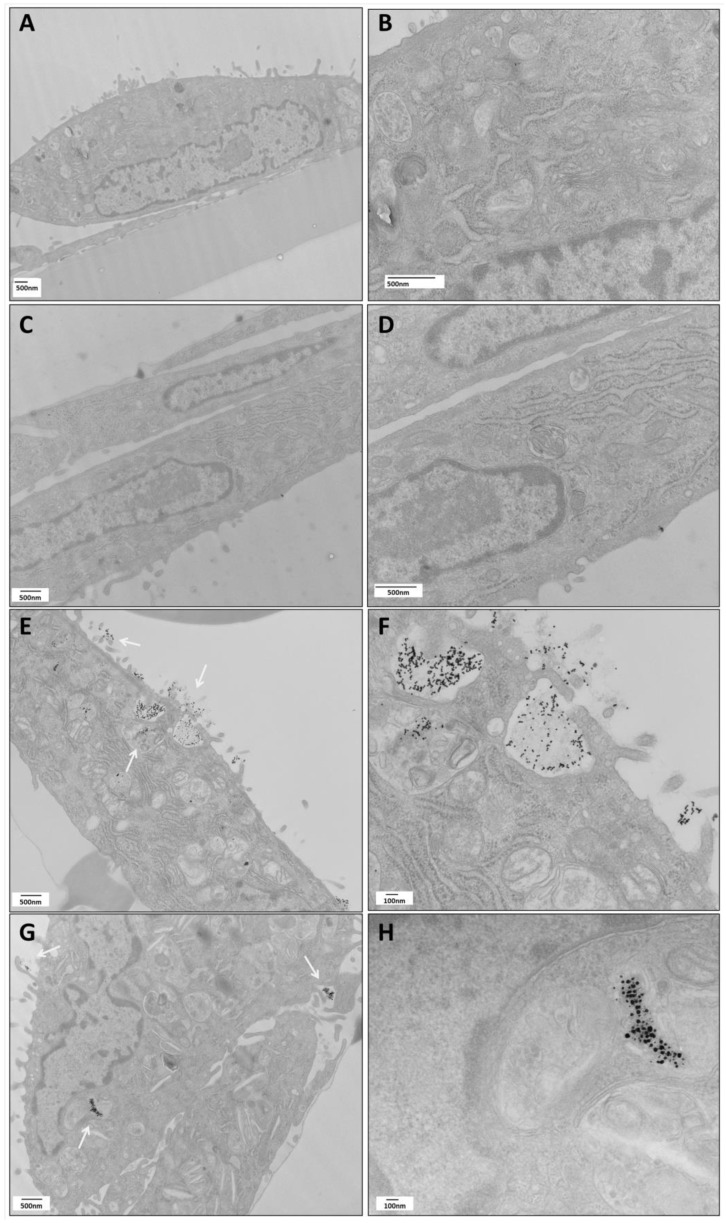
TEM images of HIEC-6 cells exposed for 24 hours to (**A**,**B**) medium alone; (**C**,**D**) PT gliadin 1 mg/mL; (**E**,**F**) AuNPs 25 ug/mL + PT1 mg/mL; (**G**,**H**) AgNPs 5 ug/mL + PT 1 mg/mL. Scale bars are reported on each panel. Arrows in panel (**E**) and (**G**) indicate mNPs outside the cells and inside autophagic vacuoles. Images were obtained with a HITACHI H-7500 TEM after proper fixation.

**Figure 2 ijms-24-13040-f002:**
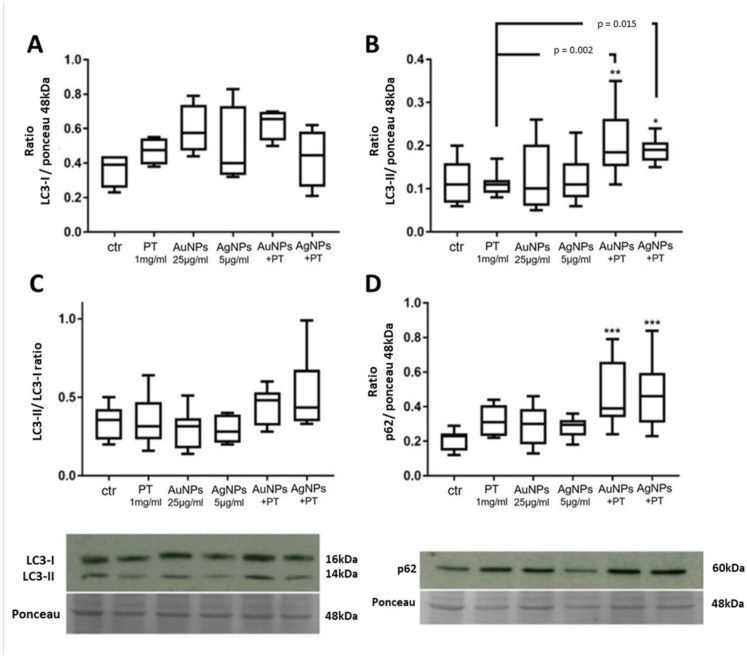
Protein quantification with Western blot assays of LC3-I (**A**), LC3-II (**B**), LC3-II/LC3-I ratio (**C**) and p62 (**D**). Target proteins were normalized with Ponceau staining in correspondence with the 48 kDa band. Significance over the ctr: * *p* < 0.05, ** *p* < 0.01, *** *p* < 0.001; significance over PT is represented with lines. Box plots represent the median, 25th and 75th percentiles. Whiskers indicate the 5th and 95th percentiles. Images are representative of at least three different experiments.

**Figure 3 ijms-24-13040-f003:**
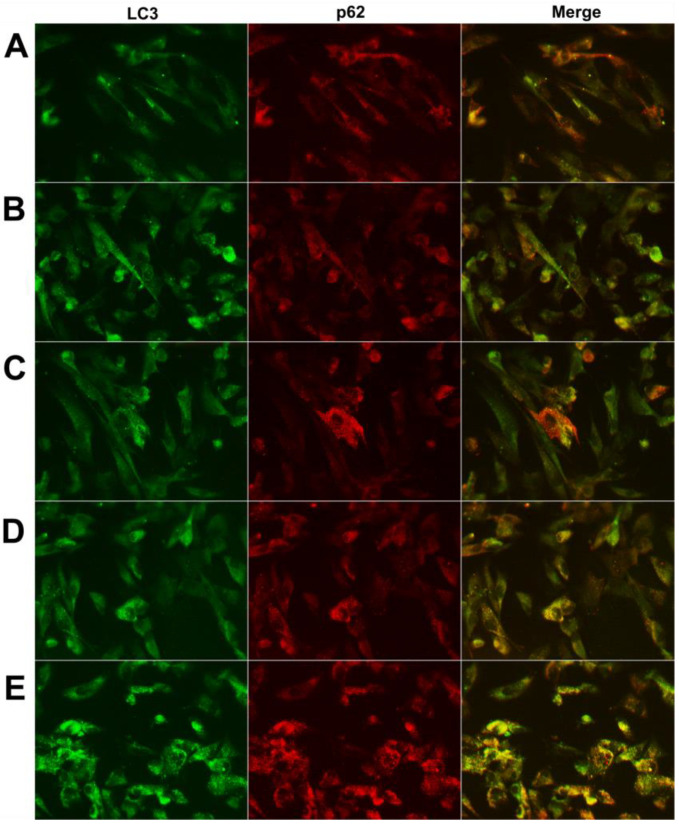
Indirect immunofluorescence of HIEC-6 cells after 24 hour treatment with (**A**) medium alone (ctr); (**B**) PT gliadin 1 mg/mL; (**C**) AuNPs 25 μg/mL + PT1 mg/mL; (**D**) AgNPs 5 μg/mL + PT 1 mg/mL; (**E**) bafilomycin 50 nM for 2 h. The green signal shows LC3; the red signal p62; the yellow signal shows the merger between the two. Images are representative of at least three different experiments. (40×).

**Figure 4 ijms-24-13040-f004:**
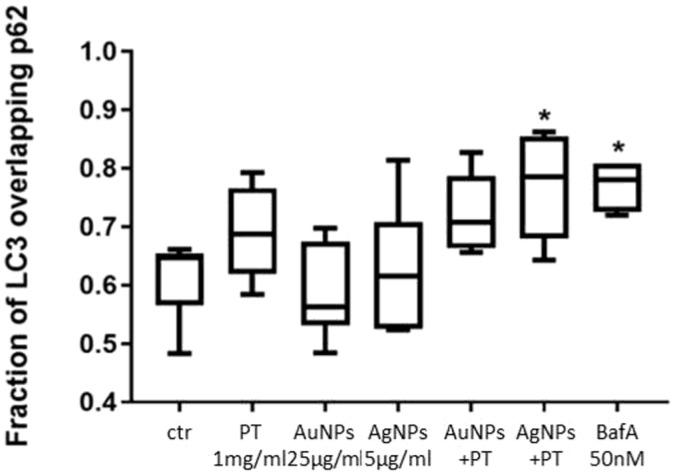
Quantification of the fraction of LC3 overlapping p62 in the indirect immunofluorescence images. Significance over the ctr: * *p* < 0.05,. Box plots represent the median, 25th and 75th percentiles. Whiskers indicate the 5th and 95th percentiles.

**Figure 5 ijms-24-13040-f005:**
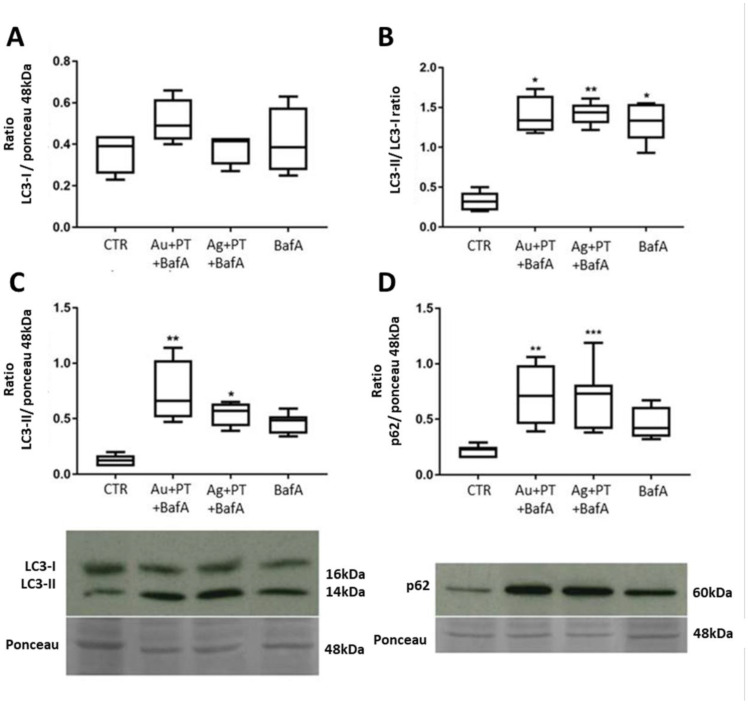
Protein quantification with Western blot assays of LC3-I (**A**), LC3-II/LC3-I ratio (**B**), LC3-II (**C**) and p62 (**D**). Target proteins were normalized with Ponceau staining in correspondence with the 48 kda band. Significance: * *p* < 0.05, ** *p* < 0.01, *** *p* < 0.001. Box plots represent the median, 25th and 75th percentiles. Whiskers indicate the 5th and 95th percentiles. Images are representative of at least three different experiments.

**Figure 6 ijms-24-13040-f006:**
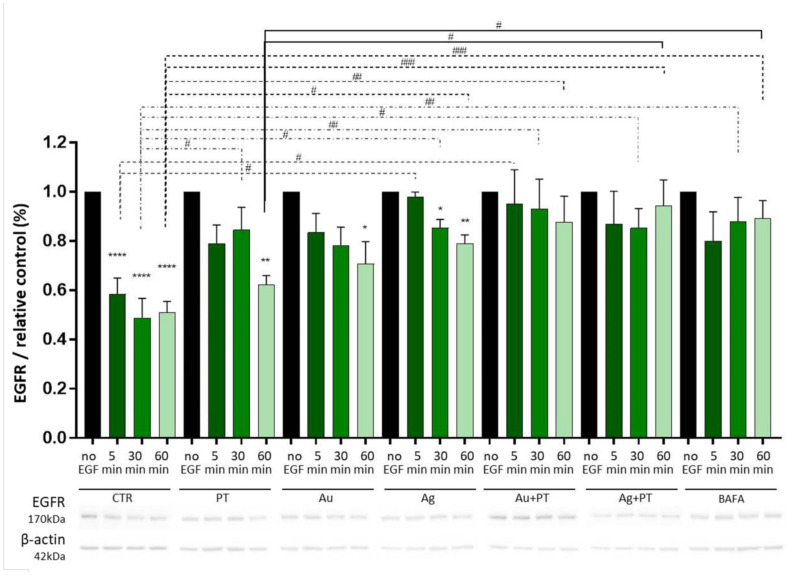
EGFR protein degradation was quantified with Western blot analysis. Before EGF pulsing, cells were treated for 24 hours with CTR = medium alone; PT = PT gliadin 1 mg/mL; Au = AuNPs 25 μg/mL; Ag = AgNPs 5 μg/mL; Au + PT = AuNPs 25 μg/mL + PT gliadin 1 mg/mL; Ag + PT = AgNPs 5 μg/mL + PT 1 mg/mL. BAFA = bafilomycin A 50 nM treatments for 2 h. EGFR blot quantification was normalized with β-actin protein. Significance of each treatment on its control: * *p* < 0.05, ** *p* < 0.01, **** *p* < 0.0001; significance among different treatments is shown with lines: # *p* < 0.05, ## *p* < 0.01, ### *p* < 0.001. Images are representative of at least three different experiments.

**Figure 7 ijms-24-13040-f007:**
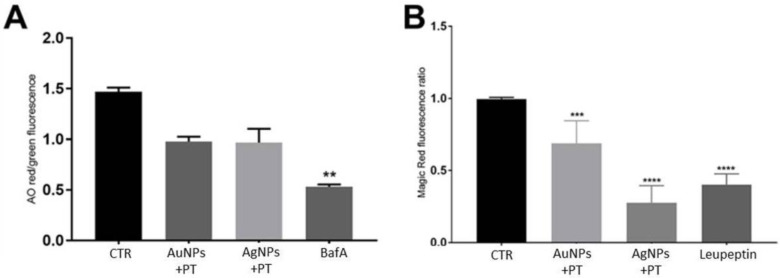
Evaluation of lysosome functionality: (**A**) evaluation of lysosome acidity through acridine orange staining; (**B**) evaluation of enzyme activity through Magic Red staining. Significance over the ctr: ** *p* < 0.01, *** *p* < 0.001, **** *p* < 0.0001.

## Data Availability

Not applicable.

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
