# Peer review of "The Combination of Gold and Silver Food Nanoparticles with Gluten Peptides Alters the Autophagic Pathway in Intestinal Crypt-like Cells"

_ijms, 2023, doi:10.3390/ijms241713040_

Round 1
Reviewer 1 Report
In the work devoted to the study of the effect of nanoparticles on autophagy in crypt-like cells of the intestine, information about the nanoparticles used is completely absent, with the exception of the manufacturer and size, apparently indicated by the manufacturer. And why is this information is included in section 4.2. HIEC-6 cell culture???
It is not clear in what form the nanoparticles were used - suspension or powder? If suspension, how was it stabilized? Powder particles tend to aggregate during storage and, accordingly, may not correspond to the declared size, and therefore I recommend including a characterization and description of the nanomaterials used.
In the methodological part, the authors do not provide a method for introducing nanoparticles into cell culture. Have suspensions been prepared? Have they been sonicated? Etc. What concentrations were used?
Judging by the figures, the working concentrations for gold were taken at 25 µg/l, for silver 5 µg/l, why exactly these? Why different? It needs to be explained.
In general, the methodological part needs to be improved. It is necessary to describe in detail how the cells were cultured (culture flasks/plates), how the nanoparticles and gliadin were introduced, at what concentrations, etc.
Abbreviations such as peptic tryptic-digested gliadin (PT) would be more appropriate to introduce at the beginning of the manuscript, as they are already found in the text and on the graphs, and are deciphered only at the end.
The description for Figure 1 says that the detected clusters are nanoparticles, was this somehow confirmed by physical and chemical analysis or just an assumption?
The scales in Figure 1 are not readable, which does not allow estimating the size of particles from the detected clusters.
Reviewer 2 Report
This is a very interesting study where the authors study how the presence of gold and silver particles with gluten peptides influences the autophagy mechanisms at the intestinal level.
Some issues are not resolved in this manuscript and it would be advisable to clarify:
1. Why do the authors use gold and silver as models of food-associated nanoparticles? Is the presence of these in the diet normal?
2. It is not clear how the authors know that gliadin + mNPs complexes are actually bound and can block autophagy. How do you measure this union?
3. In the introduction, it is necessary to broaden the focus on mNPs applications in food, contemplated by the authors in reference 12-15.
4. Avoid excessive use of abbreviations in subsections.
5. Minimally characterize the cell type used in the HIEC-6 study.
6. Why are different concentrations of AuNP and AgNP used in the study?
7. line 100. BafA, indicate abbreviation the first time it is cited.
8. Fig1. C and D, the authors should describe in more detail the differences found between both images. In fig1D, no extracellular particles appear. Could this be because their concentration in the experiment is lower?
9. Fig 2, 5, 6. The units of measurement of the ordinate axis are missing.
10. Line 132. A significant increase…….compared with?
11. Bafilomycir, its purpose should be explained.
12. ACTB? Line 391
Round 2
Reviewer 1 Report
-